# You Only Scan Once: Efficient Multi-dimension Sequential Modeling with LightNet

**Zhen Qin**                                                    *zhenqin950102@gmail.com*
*OpenNLPLab*

**Yuxin Mao**                                                   *maoyuxin@main.nwpu.edu.cn*
*Northwestern Polytechnical University*

**Xuyang Shen**                                                 *xuyangshen1122@gmail.com*
*OpenNLPLab*

**Dong Li**                                                     *liddalidd@gmail.com*
*OpenNLPLab*

**Jing Zhang**                                                  *zjnwpu@gmail.com*
*Australian National University*

**Yuchao Dai**                                                  *daiyuchao@nwpu.edu.cn*
*Northwestern Polytechnical University*

**Yiran Zhong**[*]                                              *zhongyiran@gmail.com*
*Tongyi Lab, Alibaba Group*

**Reviewed on OpenReview:** *https://openreview.net/forum?id=XG9ngiTupe*

## Abstract

Linear attention mechanisms have gained prominence in causal language models due to their linear computational complexity and enhanced speed. However, the inherent decay mechanism in linear attention presents challenges when applied to multi-dimensional sequence modeling tasks, such as image processing and multi-modal learning. In these scenarios, the utilization of sequential scanning to establish a global receptive field necessitates multiple scans for multi-dimensional data, thereby leading to inefficiencies. This paper identifies the inefficiency caused by a "multiplicative decay" linear recurrence and proposes an efficient alternative "additive decay" linear recurrence to avoid the issue, as it can handle multi-dimensional data within a single scan. We further develop an efficient multi-dimensional sequential modeling framework called LightNet based on the new recurrence. Moreover, we present two new multi-dimensional linear relative positional encoding methods, MD-TPE and MD-LRPE to enhance the model's ability to discern positional information in multi-dimensional scenarios. Our empirical evaluations across various tasks, including image classification, image generation, bidirectional language modeling, and autoregressive language modeling, demonstrate the efficacy of LightNet, showcasing its potential as a versatile and efficient solution for multi-dimensional sequential modeling.

## 1 Introduction

Linear attention has emerged as an effective alternative to softmax attention due to its linear computational complexity and enhanced processing speed, especially in causal language models (Peng et al., 2024; Qin et al.,

---

[*]Indicates the corresponding author.

2023a). The benefits of linear attention largely depend on its decay mechanism (Peng et al., 2024; Qin et al., 2023a; Sun et al., 2023b; Mao et al., 2025), which prevents attention dilution (Qin et al., 2022) and facilitates global interaction among tokens. However, the decay mechanism presents two primary issues: First, the decay mechanism is not easily applicable to high-dimensional inputs due to the need for multiple sequential scans to establish a global multi-dimensional receptive field, which reduces computational efficiency (Duan et al., 2024; Zhu et al., 2024). Additionally, without the decay mechanism, linear attention lacks positional awareness during computations, leading to decreased performance (Qin et al., 2022). In light of these challenges, we are investigating the feasibility of reducing sequential scans for multi-dimensional scenarios while preserving performance.

We first analyze the types of linear recurrence and divide them into two categories: *multiplicative* and *additive*. In multiplicative recurrence, the decay rate is dependent only on the current moment, making it impossible to obtain information about subsequent moments with a single scan. By taking image processing as an example, using multiplicative decay recurrence will require at least two scans to retrieve the global information (Duan et al., 2024; Zhu et al., 2024; Mao et al., 2026b). Conversely, in additive decay recurrence, the decay rate depends on all moments through the summation of the importance score of each moment, enabling it to gather global information in a single scan.

It is important to note that in non-causal situations, additive recurrence is permutation-invariant, which means it lacks local precedence and therefore diminishes the capture of positional information. To overcome this limitation, we put forth a new approach to positional encoding called Multi-Dimensional Toeplitz Positional Encoding (MD-TPE). This method utilizes the mathematical properties of the Toeplitz matrix to embed relative positional information with linear time complexity, thus ensuring efficiency in multi-dimensional scenarios. Additionally, we expand the Linearized Relative Positional Encoding (LRPE) (Qin et al., 2023b) to high-dimensional scenarios, resulting in the creation of Multi-Dimensional Linearized Relative Positional Encoding (MD-LRPE).

We then present LightNet, a new multi-dimensional linear attention model built on additive decay recurrence. LightNet features a pioneering decay mechanism, allowing for efficient single-scan processing of high-dimensional sequential data. Furthermore, it integrates highly effective multi-dimensional position encoding such as MD-TPE and MD-LRPE to precisely capture positional information. LightNet achieves speed advantage in non-causal settings while maintaining comparable speed in causal settings.

We conduct several evaluations of the performance of our proposed LightNet on a range of tasks, including image generation, image classification, bidirectional language modeling, and autoregressive language modeling. LightNet performs comparably or better than its competitors across all tasks.

We summarize our main contributions as follows:

- We analyze the types of linear recurrence, dividing them into two types: *multiplicative* and *additive*, where the additive type can obtain global information in a single scan.
- We propose two multi-dimensional position encoding strategies, MD-TPE and MD-LRPE, to effectively capture positional information in multi-dimensional scenarios.
- We propose LightNet, a new multi-dimensional linear attention model that can process high-dimensional sequences in a single scan.
- We conduct thorough evaluations to assess the efficiency and efficacy of LightNet for multi-dimensional sequential modeling tasks. The LightNet demonstrates competitive performance in all scenarios.

## 2 Related Work

**Linear Attention.** The linear attention mechanism has greatly advanced deep learning, particularly in natural language processing, by providing a scalable solution for long input sequences and reducing the computational demands of traditional attention models (Choromanski et al., 2020; Katharopoulos et al., 2020; Qin et al., 2021; Mao et al., 2026a). However, despite its faster training speeds, the performance of linear attention still falls short of softmax attention due to the attention dilution issue (Qin et al., 2022). The TNL/RetNet (Qin et al., 2022; 2023a) introduces a decay mechanism to address this problem. Additionally,

GLA (Yang et al., 2023) incorporating gating mechanisms shows the potential to enhance linear attention models.

**State Space Model.** State Space Models (SSMs) are increasingly crucial in sequence modeling due to their structured approach to capturing temporal dynamics through latent variables. The S4 model (Gu et al., 2021) enhances state space modeling for long sequences by leveraging structured spaces to improve computational efficiency and tackle complex dynamics. With additional parameterizing and initializing diagonal state space strategy (Gu et al., 2022), the SSMs can achieve comparable performance to naive transformers. Furthermore, the Gated State Space (GSS) model (Mehta et al., 2023) introduces a gating mechanism to SSMs, which is particularly effective for long-range language modeling by allowing nuanced control over information flow. The S5 model (Smith et al., 2022) reduces complexity using "scan" while maintaining the capability to handle intricate sequences. However, directly extending the SSM to multi-dimensional input usually requires multiple sequential scans, which will reduce the computational efficiency (Zhu et al., 2024).

**Linear RNN.** Linear RNNs employ element-wise recursion for sequence modeling, and due to their linear recursive form, they can be accelerated using parallel scans (Martin & Cundy, 2018). At their core is the decay mechanism, where RWKV-4/LRU (Peng et al., 2024; Orvieto et al., 2023) utilizes data-independent decay. HGRN (Qin et al., 2024c;b) leverage data-dependent decay to enhance performance. Linear RNNs have shown considerable potential in language modeling and long-sequence modeling tasks.

**Multi-dimensional Tasks with Linear Complexity Model.** The development of linear attention in language models has led to its extension into multi-dimensional tasks. Building upon the cosFormer framework (Qin et al., 2021), VVT (Sun et al., 2023a) explores a local prior of 2D linear attention and applies it to image classification tasks. Vim (Zhu et al., 2024) and Vision-RWKV (Duan et al., 2024) utilize a sequential scan mechanism to expand Mamba (Gu & Dao, 2023) and RWKV (Peng et al., 2023) for image classification. Additionally, leveraging the benefits structure of the diffusion transformer (Peebles & Xie, 2023) in image generation, several works have extended linear complexity models into 2D space (Fei et al., 2024a;b; Yan et al., 2023; Hu et al., 2024) to replace the traditional transformer architecture, achieving efficient image generation. However, some of these tasks encounter issues with inadequate performance. Moreover, frequent sequential scans can compromise the efficiency of the model.

## 3 Preliminary

In this section, we provide preliminary knowledge about softmax attention (Vaswani et al., 2017), linear attention (Katharopoulos et al., 2020), and linear attention with decay (Qin et al., 2021; 2024a).

**Softmax attention** operates on query $\mathbf{Q}$, key $\mathbf{K}$ and value $\mathbf{V}$ matrices. Each of them is the image of a linear projection taking input $\mathbf{X} \in \mathbb{R}^{n \times d}$ as input:

$$\mathbf{O} = \mathrm{Softmax}(\mathbf{Q}\mathbf{K}^\top / \sqrt{d})\mathbf{V},$$

with $n$ the input length, $d$ the hidden dimension. Computing $\mathrm{Softmax}(\mathbf{Q}\mathbf{K}^\top / \sqrt{d})$ needs $O(n^2)$ time complexity, which makes Softmax attention very costly when processing long documents.

**Linear attention** removes the softmax function and uses a kernel function $\phi(.)$ (Katharopoulos et al., 2020; Qin et al., 2021; Choromanski et al., 2020) to map queries and keys to hidden representations, the formulation can be written as:

$$\mathbf{O} = \mathbf{\Delta}^{-1}\phi(\mathbf{Q})[\phi(\mathbf{K})^\top\mathbf{V}], \mathbf{\Delta} = \mathrm{diag}(\phi(\mathbf{Q})[\phi(\mathbf{K})^\top\mathbf{1}_n]).$$

Since $\phi(\mathbf{K})^\top\mathbf{V}$ is computed first, the time complexity is $O(n)$. Qin et al. (2022) find the denominator term $\mathbf{\Delta}$ makes the training unstable and replace it with an extra-normalization function, the normalization can be layernorm (Ba et al., 2016), rmsnorm (Zhang & Sennrich, 2019), srmsnorm (Qin et al., 2023a), and the formulation can be simplified as:

$$\mathbf{O} = \mathrm{Norm}\left(\phi(\mathbf{Q})[\phi(\mathbf{K})^\top\mathbf{V}]\right), \tag{1}$$

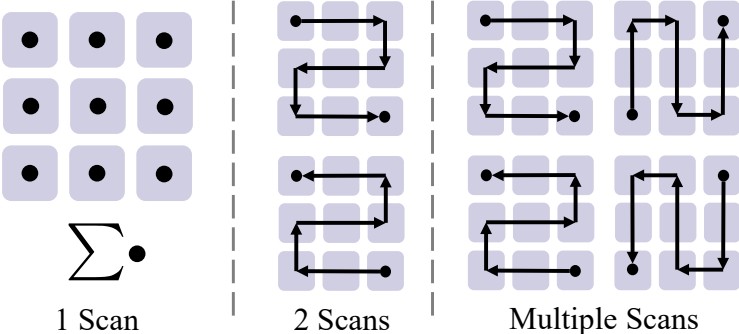

Figure 1: Illustration of different scan numbers. Different from the methods that perform multiple scans, our proposed method only performs "1 scan", which sum all tokens together directly, as shown in the figure on the left.

In a causal scenario, such as in a language model, linear attention can be written in a recursive form (Katharopoulos et al., 2020) (here we ignore normalization and kernel function $\phi$):

$$\mathbf{kv}_0 = \mathbf{0}, \mathbf{kv}_t = \mathbf{kv}_{t-1} + \mathbf{k}_t \mathbf{v}_t^\top, \mathbf{o}_t^\top = \mathbf{q}_t^\top \mathbf{kv_t}, t = 1, \ldots, n.$$

**Linear attention with decay** means that a decay term $\lambda_t$ in the recursion (Qin et al., 2023a; 2024c;b):

$$\mathbf{kv}_0 = \mathbf{0}, \mathbf{kv}_t = \lambda_t \mathbf{kv}_{t-1} + \mathbf{k}_t \mathbf{v}_t^\top, \mathbf{o}_t^\top = \mathbf{q}_t^\top \mathbf{kv_t}, t = 1, \ldots, n, 0 < \lambda_t \leq 1. \tag{2}$$

When the decay term $\lambda_t$ is independent of the input (*i.e.*, $\lambda_t = \lambda$), it is also known as data-independent decay (Qin et al., 2023a; Sun et al., 2023b). When the term $\lambda_t$ is related to the input, it is referred to as data-dependent decay (Yang et al., 2023; Qin et al., 2024c;b; Gu & Dao, 2023). Note that the decay term is essential for enhancing the performance of linear attention. Removing the decay term results in a significant drop in performance. However, the decay term also presents a major challenge when trying to effectively apply linear attention to multidimensional data as the "right product" trick cannot be used in this scenario (Qin et al., 2023a; Yang et al., 2023).

## 4 Linear Recurrence in Multi-dimensional Space

In this section, we discuss the theoretical and practical computational complexity of linear recurrence (with decay) when dealing with high-dimensional data, and then analyze the types of linear recurrence. In subsequent discussions, we assume $n$ is the sequence length, $d$ is the embedding dimension, and $\mathbf{x}_t \in \mathbb{R}^d$ is the transpose of the $t$-th row of matrix $\mathbf{X} \in \mathbb{R}^{n \times d}$.

### 4.1 Computational Complexity of Linear Recurrence

Eq. 2 illustrates the linear recurrence in causal scenarios. When dealing with non-causal scenarios, a common practice in the literature is to perform causal computation twice (Duan et al., 2024; Zhu et al., 2024). We call this method "2 scan":

$$\overrightarrow{\mathbf{kv}}_0 = \mathbf{0}, \overrightarrow{\mathbf{kv}}_t = \lambda_t \overrightarrow{\mathbf{kv}}_{t-1} + \mathbf{k}_t \mathbf{v}_t^\top, \overrightarrow{\mathbf{o}}_t^\top = \mathbf{q}_t^\top \overrightarrow{\mathbf{kv}}_t,$$
$$\overleftarrow{\mathbf{kv}}_{n+1} = \mathbf{0}, \overleftarrow{\mathbf{kv}}_t = \lambda_t \overleftarrow{\mathbf{kv}}_{t+1} + \mathbf{k}_t \mathbf{v}_t^\top, \overleftarrow{\mathbf{o}}_t^\top = \mathbf{q}_t^\top \overleftarrow{\mathbf{kv}}_t,$$
$$\mathbf{o}_t = \overrightarrow{\mathbf{o}}_t + \overleftarrow{\mathbf{o}}_t.$$

When $\lambda_t = 1$, *i.e.* there is no decay, the right product trick (Katharopoulos et al., 2020) can be applied in this case. We call this method "1 scan", as shown in Fig. 1.

$$[\mathbf{KV}] = \mathbf{K}^\top \mathbf{V}, \mathbf{O} = \mathbf{Q}[\mathbf{KV}].$$

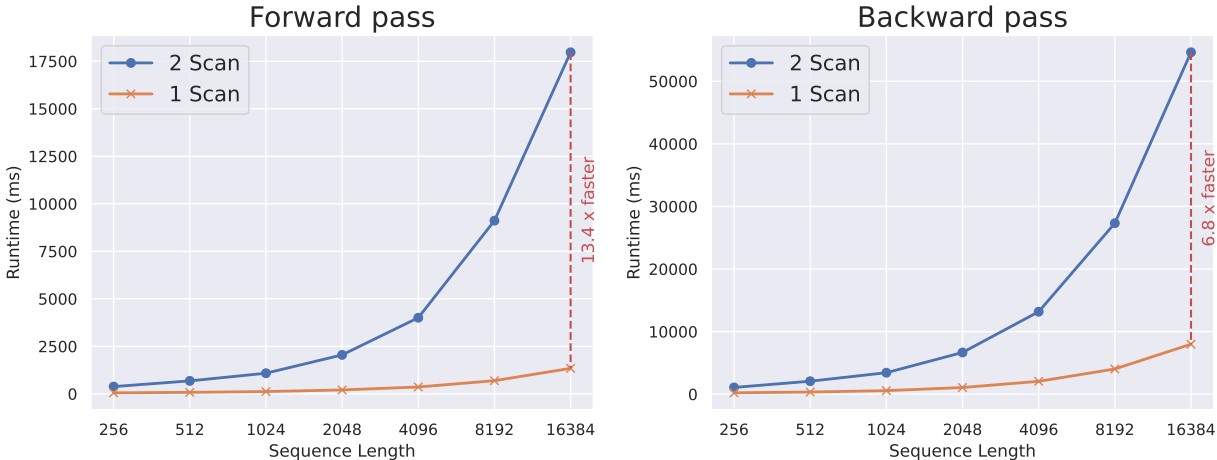

Figure 2: **Processing time of 1 Scan and 2 Scan in relation to sequence length.** 1 Scan is significantly faster than 2 Scan in both forward and backward passes. As the sequence length increases, the advantage of 1 Scan becomes more substantial. Note that the x-axis scale follows a logarithmic scale to enhance visualization clarity.

Although both of the above formulas have a time complexity of $O(nd^2)$, the "2 scan" version is significantly slower than the "1 scan" version. This is because causal computation requires block-level recursion (Qin et al., 2024a; Yang et al., 2023), whereas the second formula can be fully parallelized due to matrix multiplication (Katharopoulos et al., 2020). We provide a speed comparison in Fig. 2, where the "2 scan" is implemented with Lightning Attention (Qin et al., 2024a), the fastest linear attention implementation so far. It can be seen that the "2 scan" is several times slower than the "1 scan" in both forward and backward passes.

It is apparent that the need for multiple scans is mainly due to the presence of decay $\lambda_t$. However, directly removing $\lambda_t$ would lead to degraded performance (Qin et al., 2022). A natural question arises: *can we retain $\lambda_t$ while only performing a single scan?* In the next section, we will discuss the types of linear recurrence and answer the question.

## 4.2 Types of Linear Recurrence

We first explore the representation range of linear recurrences by 1D linear recurrence, Here, we assume $a_t \triangleq f(x_1, \ldots, x_t)$, $f : \mathbb{R} \to \mathbb{R}$ is some function. It indicates that $a_t$ is data-dependent, i.e., depending on the input tokens.[1]:

$$y_t = a_t y_{t-1} + x_t, y_0 = 0. \tag{3}$$

Unroll the recursion equation of Eq. 3, we obtain:

$$y_t = \sum_{s=1}^{t} \frac{A_s}{A_t} x_s \triangleq \sum_{s=1}^{t} c_{ts} x_s, A_t = \left( \prod_{s=1}^{t} a_s \right)^{-1}. \tag{4}$$

The detailed proof of the unrolling process can be found in Appendix A.1. Note that $y_t$ is a linear combination of $x_1, \ldots, x_t$. A natural question arises: *Can every linear combination $\sum_{s=1}^{t} c_{ts} x_s$ be represented as a linear recursion?* We now prove that a linear recursion representation is possible only when the coefficients $c_{ts}$ satisfy certain conditions.

**Theorem 4.1.** *A linear recurrence $y_t = a_t y_{t-1} + x_t, y_0 = 0$ is equivalent to a linear combination $y_t = \sum_{s=1}^{t} c_{ts} x_s$, iff $c_{ts} = \frac{g_s}{g_t}$, where $g_t = g(x_1, \ldots, x_t)$.*

---

[1]This assumption is commonly adopted in the Linear Attention and RNN communities. (Yang et al., 2023; Gu & Dao, 2023)

*Proof of Theorem 4.1.* $\Rightarrow$

Given a linear recurrence, we multiply it by $A_t = \left(\prod_{s=1}^{t} a_s\right)^{-1}$ and following recurrence equation:

$$A_t y_t = A_t a_t y_{t-1} + A_t x_t = A_{t-1} y_{t-1} + A_t x_t.$$

Unroll it, we get:

$$A_t y_t - A_{t-1} y_{t-1} = A_t x_t, \ldots, A_2 y_2 - A_1 y_1 = A_2 x_2. \tag{5}$$

To derive an expression for $y_t$, we sum the recursive equations and obtain:

$$A_t y_t - A_1 y_1 = \sum_{s=2}^{t} A_s c x_s, \quad y_t A_t = \sum_{s=1}^{t} A_s x_s, \quad y_t = \sum_{s=1}^{t} \frac{A_s}{A_t} x_s. \tag{6}$$

By comparing the coefficients, we can obtain $c_{ts} = A_s / A_t$.

$\Leftarrow$:

Given the linear combination $y_t = \sum_{s=1}^{t} c_{ts} x_s$ and $c_{ts} = \frac{g_s}{g_t}$, we define $a_t \triangleq \frac{g_{t-1}}{g_t}$. Then $y_t$ can be expressed as:

$$y_t = \sum_{s=1}^{t} c_{ts} x_s = \sum_{s=1}^{t-1} c_{ts} x_s + c_{tt} x_t = \sum_{s=1}^{t-1} \frac{g_s}{g_t} x_s + \frac{g_t}{g_t} x_t$$

$$= \frac{g_{t-1}}{g_t} \sum_{s=1}^{t-1} \frac{g_s}{g_{t-1}} x_s + x_t = a_t \sum_{s=1}^{t-1} c_{t-1,s} x_s + x_t = a_t y_{t-1} + x_t. \quad \square$$

Based on the Theorem 4.1, for linear recurrence, we can directly discuss $g_t$, as $a_t$ can be obtained through $\frac{g_{t-1}}{g_t}$. Intuitively, $g_t$ can be interpreted as an importance score up to moment $t$, $c_{ts} = \frac{g_s}{g_t}$ can be interpreted as the ratio of the score at moment $s$ relative to moment $t$, and $a_t$ can be interpreted as the ratio of the previous moment's score to moment $t$'s score.

Typically, to prevent numerical overflow, we assume $0 \le a_t = \frac{g_{t-1}}{g_t} \le 1$. To meet this condition, we present the following two forms:

**Proposition 4.2.** *For Linear Recurrence with $0 \le a_t \le 1$, there exist two forms:*
*1. Multiplicative decay:* $\log g_t = \log g_{t-1} + \delta_t, a_t = \exp(-\delta_t)$;
*2. Additive decay:* $g_t = g_{t-1} + \delta_t, a_t = \frac{\sum_{s=1}^{t-1} \delta_s}{\sum_{s=1}^{t} \delta_s}$;

*where $\delta_t \triangleq \delta(x_t) \ge 0$.*

*Proof of Proposition 4.2.* The condition $0 \le \frac{g_{t-1}}{g_t} \le 1$, is equivalent to $\delta_t = \log g_t - \log g_{t-1} \ge 0$ or $\delta_t = g_t - g_{t-1} \ge 0$. The former formula brings the multiplicative type, while the latter delivers the additive type. $\square$

It can be observed that the typical linear attention with decay corresponds to the Multiplicative decay, where $\delta_t$ is utilized as Softplus($\cdot$) (Yang et al., 2023; Gu & Dao, 2023), exp($\cdot$)) (Gu & Dao, 2023), or a fixed value (Qin et al., 2023a; Sun et al., 2023b). Since the $a_t$ in Multiplicative decay depends solely on the input $x_t$ at the current timestep, a single scan cannot enable $y_t$ to capture the information from $x_1, \ldots, x_n$ ($n$ is the sequence length), i.e., the global context, when processing high-dimensional data. However, for the Additive decay, since the computation decay is $a_t = \frac{\sum_{s=1}^{t-1} \delta_s}{\sum_{s=1}^{t} \delta_s}$, by modifying the denominator to $\Delta = \sum_{s=1}^{n} \delta_s$, global information can be obtained through $a_t = \frac{\sum_{s=1}^{t-1} \delta_s}{\Delta}$.

## 4.3 More discussion on additive decay and multiplicative decay

The principal distinction between additive decay and multiplicative decay lies in their ability to represent relative distance. Specifically, additive decay cannot explicitly encode relative distance, whereas multiplicative decay can.

Consider the linear recurrence: $y_t = a_t y_{t-1} + x_t, y_0 = 0$, with its expanded form: $y_t = \sum_{s=1}^t c_{ts} x_s$.

For multiplicative decay, the coefficient $c_{ts}$ can be written as:

$$c_{ts} = \frac{g_s}{g_t} = \exp(\log g_s - \log g_t) = \exp\left(-\sum_{j=s+1}^t a_j\right). \tag{7}$$

Moreover, it satisfies the bounds:

$$\rho_1^{t-s} \le c_{ts} \le \rho_2^{t-s}, \tag{8}$$

where

$$\rho_1 = \min_j \exp(-a_j), \rho_2 = \max_j \exp(-a_j). \tag{9}$$

Thus, the upper and lower bounds of $c_{ts}$, which can be interpreted analogously to attention scores, are explicit functions of the relative distance $t - s$.

For additive decay, the coefficient is given by:

$$c_{ts} = \frac{\sum_{i=1}^s g_i \delta_i}{\sum_{i=1}^t g_i \delta_i}. \tag{10}$$

It follows that:

$$\frac{s\mu_1}{t\mu_2} \le c_{ts} \le \frac{s\mu_2}{t\mu_1}, \tag{11}$$

where

$$\mu_1 = \min_j \delta_j, \qquad \mu_2 = \max_j \delta_j. \tag{12}$$

In contrast to the multiplicative case, the bounds of $c_{ts}$ under additive decay depend on the absolute indices $s$ and $t$, rather than solely on their relative distance $t - s$.

Therefore, multiplicative decay yields coefficients whose bounds are explicit functions of relative distance, while additive decay does not. This fundamental difference motivates the introduction of MD-TPE and MD-LRPE in our subsequent design, as they enable explicit modeling of relative positional structure.

## 5 LightNet

Building upon the preceding analysis, we introduce a novel Linear Transformer architecture termed **Light-Net**, designed to handle multi-dimensional data efficiently in 1 scan. An overview of its structure is depicted in Fig. 3. LightNet comprises an Input Embedding, MD-TPE module, and several stacked LightNet Layers.

### 5.1 LightNet Layer

The LightNet Layer is composed of a LightNet Attention (LNA) and a Gated Linear Unit (GLU) (Shazeer, 2020). Within the LNA, an additive decay is employed, with $\delta$ implemented through the exponential function. Additionally, a parameter sharing strategy (Qin et al., 2024b) is utilized for both the key and decay, which has been empirically observed to enhance performance. This empirical evidence is detailed in Table 5. Furthermore, the integration of a low-rank output gate from TNL3 (Qin et al., 2023a) and a normalization after linear attention (Qin et al., 2022) has been incorporated.

In causal settings, the LightNet Layer can be represented as follows:

$$\mathbf{s}_t = \mathbf{s}_{t-1} + \exp(\mathbf{k}_t), \bar{\mathbf{k}}_\mathbf{t} = \exp(\mathbf{k}_t)/\mathbf{s}_t, \mathbf{kv}_t = \mathrm{diag}\{1 - \bar{\mathbf{k}}_\mathbf{t}\}\mathbf{kv}_{t-1} + \bar{\mathbf{k}}_\mathbf{t}\mathbf{v}_t^\top,$$
$$\mathbf{o}_t^\top = \mathrm{Norm}[\mathbf{kv}_t^\top \phi(\mathbf{q}_t)] \odot \psi(u_t). \tag{13}$$

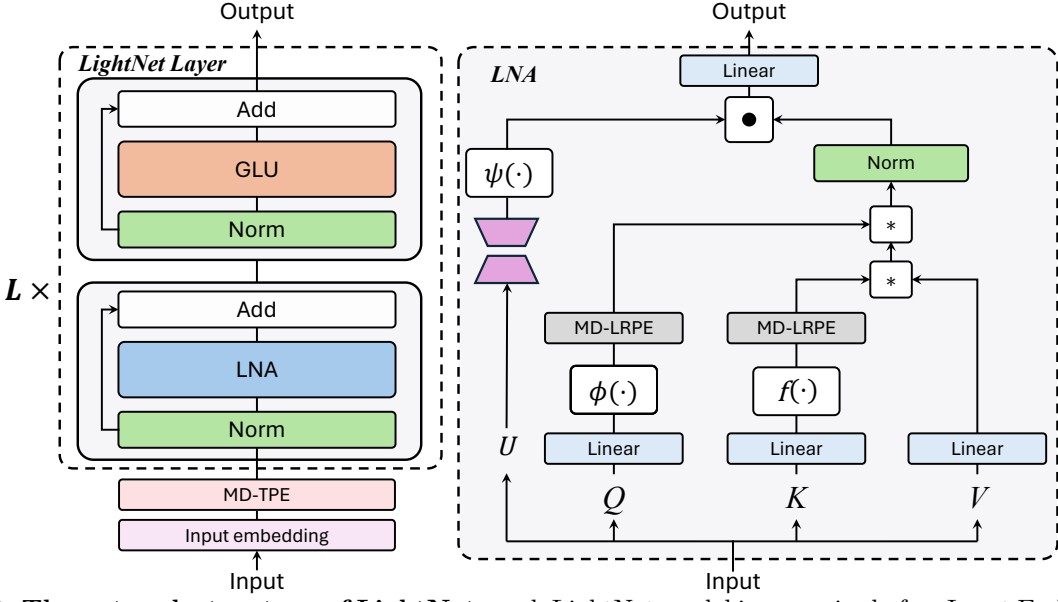

Figure 3: **The network structure of LightNet**: each LightNet model is comprised of an Input Embedding, MD-TPE, and a stack of multiple LightNet Layers. Each LightNet Layer consists of an LNA(LightNet Attention) and a GLU, with the computation of LNA illustrated in the figure on the right.

In non-causal settings, the expression becomes:

$$\mathbf{s} = \sum_t \exp(\mathbf{k}_t), \mathbf{o}_t = \text{Norm}\left[\phi(q_t)\sum_t (\exp(k_t)/s)^\top \mathbf{v}_t\right] \odot \psi(u_t),$$
$$\mathbf{O} = \text{Norm}\left[\phi(\mathbf{Q})(f(\mathbf{K})^\top \mathbf{V})\right] \odot \psi(\mathbf{U}).$$

(14)

where $\mathbf{X}$ is the input of LNA, $\mathbf{W}_q, \mathbf{W}_k, \mathbf{W}_v$ are linear projection matrices and $\mathbf{W}_{u1}, \mathbf{W}_{u2}$ are low rank projection of output gates:

$$\mathbf{Q} = \mathbf{X}\mathbf{W}_q, \mathbf{K} = \mathbf{X}\mathbf{W}_k, \mathbf{V} = \mathbf{X}\mathbf{W}_v, \mathbf{U} = \mathbf{X}\mathbf{W}_{u1}\mathbf{W}_{u2}, \phi = \text{Swish}, \psi = \text{Sigmoid}, f = \text{Softmax}.$$

(15)

### 5.2 Multi Dimension Position Encoding

It is noted that additive decay recurrence does not have a locality prior like multiplicative decay recurrence and is permutation invariant in non-causal scenarios, as shown in E.q 14. Therefore, it is necessary to introduce new positional encoding. We choose to use relative positional encoding due to its superior performance compared to absolute positional encoding (Shaw et al., 2018). However, existing relative positional encoding methods for Transformers are incompatible with LightNet, as they either require direct manipulation of the attention scores (Shaw et al., 2018) or fail to retain the benefits of relative positional information (Su et al., 2021). For detailed discussions, see Appendix A.2. This necessitates designing a positional encoding scheme tailored for LightNet. To tackle this challenge, we introduce two novel relative positional encoding methods, MD-TPE (Multi-Dimensional Toeplitz Positional Encoding) and MD-LRPE (Multi-Dimensional Linearized Relative Positional Encoding), which is the high-dimensional context expanding of the LRPE (Qin et al., 2023b). This expanding of MD-LRPE enables the management of relative positional relationships in any dimension.

**MD-TPE.** Given multi-dimension input $\mathbf{x}_{n_1,\ldots,n_k}, 1 \le n_s \le N_s, s = 1,\ldots k$, we use the following equation to capture relative positional information:

$$\mathbf{y}_{n_1,\ldots,n_k} = \sum_{m_k \le n_k} \cdots \sum_{m_1 \le n_1} \mathbf{t}_{n_1-m_1,\ldots,n_k-m_k}\mathbf{x}_{m_1,\ldots,m_k}.$$

(16)

However, the time complexity of implementing the aforementioned method is $O(N \log N)$, where $N = \prod_{s=1}^{k} n_s$, making it inefficient. To address this, we simplified the above formula by performing toeplitz matrix production for

Table 1: **Performance comparison for image classification task on ImageNet1k.** "S.A." represents Softmax Attention, "M.S." denotes multiple scans, and "O.S." signifies one scan. The best result is highlighted with **bold** and the second with underlined.

| Model | Category | Tiny | | Small | | Base | |
|---|---|---|---|---|---|---|---|
| | | Acc (%) ↑ | Params (M) | Acc (%) ↑ | Params (M) | Acc (%) ↑ | Params (M) |
| DeiT (Touvron et al., 2021) | S.A. | 72.20 | 5.7 | 79.90 | 22.00 | 81.80 | 86.00 |
| Hgrn (Qin et al., 2024c) | M.S. | 74.40 | 6.1 | 80.09 | 23.70 | - | - |
| Vim (Zhu et al., 2024) | M.S. | **76.10** | 7.0 | **80.50** | 26.00 | - | - |
| V-RWKV (Duan et al., 2024) | M.S. | 75.10 | 6.2 | 80.10 | 23.80 | **82.00** | 93.70 |
| Tnl (RetNet) (Qin et al., 2023a) | M.S. | 72.89 | 6.0 | 78.76 | 22.56 | 80.62 | 87.59 |
| Hgrn2 (Qin et al., 2024b) | M.S. | 75.39 | 6.1 | 80.12 | 23.80 | - | - |
| LightNet | O.S. | 74.46 | 6.0 | 80.12 | 22.64 | 81.90 | 87.74 |
| LightNet w/o TPE | O.S. | 73.97 | 6.0 | 79.65 | 22.54 | 81.45 | 87.54 |
| LightNet w/o LRPE | O.S. | 74.02 | 6.0 | 79.54 | 22.63 | 81.72 | 87.69 |
| LightNet w/o Decay | O.S. | 71.85 | 6.0 | 79.95 | 22.64 | 80.71 | 87.74 |

each dimension separately and using SSM for parameterization (Qin & Zhong, 2023; Gu et al., 2021; Ma et al., 2023; 2024), we denote $e$ as the hidden dimension of SSM below:

$$\mathbf{y}_{n_1,\ldots,n_k} = \sum_{s=1}^{k}\sum_{m_s=1}^{n_s}\mathbf{t}_{n_s-m_s}\mathbf{x}_{n_1,\ldots,m_s,\ldots,n_k} = \sum_{s=1}^{k}\sum_{m_s=1}^{n_s}\sum_{r=1}^{e}\lambda_r^{n_s-m_s}\mathbf{x}_{n_1,\ldots,m_s,\ldots,n_k}. \tag{17}$$

Where $\lambda_r$ is decay factor for t-th feature of SSM. By using a scan approach, the above calculation becomes linear in complexity, $O(Ne)$.

**MD-LRPE.** Given $\mathbf{x}_t \in \mathbb{R}^d, \mathbf{x} \in \{\mathbf{q}, \mathbf{k}\}$, LRPE transforms it through the matrix $\mathbf{W}_t$ to $\mathbf{W}_t\mathbf{x}_t, \mathbf{x} \in \{\mathbf{q}, \mathbf{k}\}$, and it holds that:

$$(\mathbf{W}_s\mathbf{q}_s)^{\mathrm{H}}(\mathbf{W}_t\mathbf{k}_t) = \mathbf{q}_s^{\mathrm{H}}\mathbf{W}_s^{\mathrm{H}}\mathbf{W}_t\mathbf{k}_t = \mathbf{q}_s^{\mathrm{H}}\mathbf{W}_{t-s}\mathbf{k}_t. \tag{18}$$

We choose the complex version of LRPE, where:

$$\mathbf{W}_t = \mathrm{diag}\{\exp(it\theta_1),\ldots,\exp(it\theta_d)\}. \tag{19}$$

To generalize to higher dimensions, *i.e.*, given $\mathbf{x}_{n_1,\ldots,n_k} \in \mathbb{R}^d, \mathbf{x} \in \{\mathbf{q}, \mathbf{k}\}$, we divide the $d$ features into $k$ groups, each group has $d/k$ features, with the $s$-th group's features corresponding to dimension $n_s, s \in [1, k]$. Specifically, we define:

$$\mathbf{W}_{n_1,\ldots,n_k} = \mathrm{diag}\{[\Theta_1,\ldots,\Theta_k]\}, \Theta_s = \exp(in_k\theta_j), sd/k < j \leq (s+1)d/k, \theta_j = 10000^{-2j/d}. \tag{20}$$

Thus:
$$\mathbf{W}_{n_1,\ldots,n_k}^{\mathrm{H}}\mathbf{W}_{m_1,\ldots,m_k} = \mathbf{W}_{m_1-n_1,\ldots,m_k-n_k} \tag{21}$$

Then:
$$(\mathbf{W}_{n_1,\ldots,n_k}\mathbf{q}_{n_1,\ldots,n_k})^{\mathrm{H}}(\mathbf{W}_{m_1,\ldots,m_k}\mathbf{k}_{m_1,\ldots,m_k}) = \mathbf{q}_{n_1,\ldots,n_k}^{\mathrm{H}}\mathbf{W}_s^{\mathrm{H}}\mathbf{W}_t\mathbf{k}_{m_1,\ldots,m_k}$$
$$= \mathbf{q}_{n_1,\ldots,n_k}^{\mathrm{H}}\mathbf{W}_{m_1-n_1,\ldots,m_k-n_k}\mathbf{k}_{m_1,\ldots,m_k}. \tag{22}$$

From a practical standpoint, MD-TPE is applied only once, immediately after the embedding layer. In contrast, MD-LRPE is incorporated into every attention module and is therefore applied as many times as the number of transformer layers.

Consequently, MD-TPE incurs lower computational overhead than MD-LRPE. In scenarios where maximizing computational efficiency is critical, we recommend using MD-TPE exclusively.

From an inductive bias perspective, MD-TPE introduces relative positional information via exponential decay, thereby imposing a strong locality prior. In contrast, MD-LRPE does not rely on exponential decay and thus embodies a comparatively weaker locality prior.

## 6 Experiments

We comprehensively evaluate the substitutability of our LightNet in performance, scalability, flexibility, and efficiency. We validate the effectiveness of our model on various multi-dimensional sequential modeling tasks. We also test the proposed ability of LightNet to serve as a language model.

Table 2: **Performance Scores on GLUE Benchmark.** We utilize the Cramming-BERT 24-hour training configuration and observe that LightNet outperforms Crammed BERT and achieves comparable results to BERT-Base, which is trained with more GPU hours. The best result is highlighted with **bold** and the second with underlined.

| Model | MNLI | SST-2 | STSB | RTE | QNLI | QQP | MRPC | CoLA | GLUE |
|---|---|---|---|---|---|---|---|---|---|
| BERT-Base (Fully trained) | 83.2 / 83.4 | 91.9 | **86.7** | **59.2** | **90.6** | **87.7** | **89.3** | **56.5** | **80.9** |
| BERT-Base (No Pretrain) | 34.1 / 34.1 | 79.9 | 17.8 | 47.3 | 50.0 | 68.6 | 77.9 | - | 45.5 |
| Crammed BERT | **83.9** / **84.1** | 92.2 | 84.6 | 53.8 | 89.5 | 87.3 | 87.5 | 44.5 | 78.6 |
| LightNet | 83.3 / 83.5 | **92.9** | 86.3 | 55.6 | 89.1 | **87.7** | 88.5 | 52.6 | 79.9 |
| LightNet w/o TPE | 82.1 / 82.9 | 92.4 | 79.4 | 57.8 | 89.2 | **87.7** | 83.8 | 44.1 | 77.7 |
| LightNet w/o LRPE | 82.0 / 82.7 | 92.7 | 76.3 | 57.4 | 88.5 | 87.5 | 83.8 | 38.2 | 76.6 |

## 6.1 Setting

**Image Classification.** We trained our LightNet model for image classification on the ImageNet-1K dataset (Deng et al., 2009). Our approach modifies the network architecture and training protocols of DeiT (Touvron et al., 2021), substituting its Transformer Layers with our proprietary LightNet Layers.

**Image Generation.** We build our model upon the latent diffusion model (Rombach et al., 2022; Peebles & Xie, 2023) and use our proposed LightNet as the denoising network. We adjust the model size across various configurations (S, B, L, XL) and patch sizes (8, 4, 2), consistent with DiT (Peebles & Xie, 2023). Experiments are conducted on the ImageNet dataset (Deng et al., 2009) at a resolution of $256 \times 256$. We compare the performance with typical methods for image generation, CDM (Ho et al., 2022), LDM (Rombach et al., 2022), and DiT (Peebles & Xie, 2023). Each model is trained over 0.4M steps with a batch size of 256 to assess scaling capabilities. For the largest model variant, training is extended to 0.8M steps with a batch size of 1024, as opposed to the 7M steps in DiT, to enhance generative performance.

Table 3: **Performance comparison for image generation task on ImageNet-1k.** LightNet-XL/2 achieves state-of-the-art FID with or without classifier-free guidance (-G).The best result is highlighted with **bold** and the second with underlined.

| Model | FID↓ | sFID↓ | IS↑ | Precision↑ | Recall↑ | Params |
|---|---|---|---|---|---|---|
| CDM | 4.88 | - | 158.71 | - | - | - |
| LDM-8 | 15.51 | - | 79.03 | 0.65 | 0.63 | 395M |
| LDM-8-G | 7.76 | - | 209.52 | 0.84 | 0.35 | 506M |
| LDM-4 | 10.56 | - | 103.49 | 0.71 | 0.62 | 400M |
| LDM-4-G | 3.60 | - | 247.67 | **0.87** | 0.48 | 400M |
| DiT-XL/2 | 9.62 | 6.85 | 121.50 | 0.67 | **0.67** | 675M |
| DiT-XL/2-G | 2.27 | 4.60 | 278.24 | 0.83 | 0.57 | 675M |
| LightNet-XL/2 | 5.35 | 5.93 | 171.18 | 0.73 | 0.65 | 672M |
| LightNet-XL/2-G | **2.18** | **4.58** | **281.85** | 0.83 | 0.58 | 672M |

**Bidirectional Language Modeling.** We utilize Cramming-BERT (Geiping & Goldstein, 2022) as our pipeline, employing a 24-hour training regime to pre-train on the Pile dataset, subsequently finetuning on the GLUE benchmark (Wang et al., 2018). During pre-training, we follow established guidelines by setting a learning rate of 1e-3, a sequence length of 128, and a batch size of 8192. In the finetuning phase, we experiment with learning rates from the set {5e-5, 4e-5, 3e-5, 2e-5} and determine the optimal outcome by finetuning over 5 epochs.

**Autoregressive Language Modeling.** We evaluate two capabilities: perplexity (PPL) and zero-shot reasoning ability. The perplexity of the 44M model is assessed on the Wikitext-103 dataset (Merity et al., 2016), and the 380M model's perplexity is tested on the Pile dataset, consuming 10 billion tokens . For large language model experiments, we train LightNet models at scales of 1B, and 3B using 300 billion tokens sampled from subsets of the Pile (Gao et al., 2020). These models are then evaluated on commonsense reasoning tasks using the lm-eval-harness (Gao et al., 2021). Detailed training hyperparameters are listed in Table 12.

**Efficiency Benchmark.** We benchmark the speed of LightNet under both causal and non-causal settings. In the non-causal setting, we compare the speed of the 1-Scan and 2-Scan operators, with the results shown in Fig. 2. These results indicate that the efficiency advantages of LightNet are primarily observed in the non-causal case. In the causal setting, we compare the throughput of LightNet against GLA, HGRN2, and Mamba2 on language modeling tasks, with the results reported in Table 4. In contrast to the non-causal case, LightNet shows comparable throughput in

Table 4: **Throughput comparison of different sequence modeling architectures in the causal setting.** All models have approximately the same parameter scale of 1.45B. TGS denotes the number of tokens processed per GPU per second. Compared with HGRN2, GLA, and Mamba2, LightNet is approximately 7% slower in the causal setting.

| Model | Params | TGS | Model | Params | TGS |
|---|---|---|---|---|---|
| HGRN2 | 1.45B | 18386.1 | GLA | 1.45B | 18354.5 |
| Mamba2 | 1.45B | 18428.3 | LightNet | 1.45B | 16992.2 |

the causal setting, but is slightly behind the alternatives, running approximately 7% slower than HGRN2, GLA, and Mamba2.

## 6.2 Results

**Image Classification.** As shown in Table 1, the proposed LightNet shows competitive performance on the ImageNet-1k dataset. It can be observed that using only a single sequential scan, LightNet can achieve comparable performance to models with naive attention and multiple sequential scans.

**Image Generation.** The image generation results are presented in Table 3. Our proposed LightNet demonstrates superior performance, achieving a lower Fréchet Inception Distance (FID) and a higher Inception Score (IS) than DiT (Peebles & Xie, 2023) with fewer training steps (0.8M steps vs 7M steps). Additionally, LightNet exhibits commendable scaling capabilities, as illustrated in Fig. 4.

**Bidirectional Language Modeling.** As shown in Table 2, LightNet outperforms Crammed Bert (Geiping & Goldstein, 2022) on the GLUE dataset, demonstrating its superior capability in handling natural language understanding tasks. Despite BERT-Base (Devlin et al., 2019) achieving comparable performance, it is noteworthy that LightNet does so with a significantly lower computational cost, having been trained on a single A100 for 24 hours.

**Autoregressive Language Modeling.** In the Wikitext-103 dataset, as depicted in Table 5, LightNet surpasses all competitors on both the validation and test

Table 5: **Performance comparison on Wikitext-103**. ↓ means *lower is better*. We adopted the configuration of HGRN for Wikitext-103, and we can observe that LightNet significantly outperforms all other methods. The best result is highlighted with **bold** and the second with underlined.

| Model | PPL (val) ↓ | PPL (test) ↓ | Params (M) |
|---|---|---|---|
| *Attn-based* | | | |
| Transformer | 24.40 | 24.78 | 44.65 |
| FLASH | 25.92 | 26.70 | 42.17 |
| 1+elu | 27.44 | 28.05 | 44.65 |
| Performer | 62.50 | 63.16 | 44.65 |
| cosFormer | 26.53 | 27.06 | 44.65 |
| *RNN-based* | | | |
| S4 | 38.34 | 39.66 | 45.69 |
| DSS | 39.39 | 41.07 | 45.73 |
| GSS | 29.61 | 30.74 | 43.84 |
| RWKV-4 | 24.31 | 25.07 | 46.23 |
| LRU | 29.86 | 31.12 | 46.24 |
| HGRN | 24.14 | 24.82 | 46.25 |
| *FFT-based* | | | |
| TNN | 23.98 | 24.67 | 48.68 |
| LightNet | **23.09** | **23.75** | 45.07 |

datasets. Regarding large-scale datasets, as illustrated in Table 6, LightNet exhibits superior perplexity (PPL) compared to LLaMA (Touvron et al., 2023) and TNL (Qin et al., 2023a), and matches the performance of Mamba (Gu & Dao, 2023). The ability of LightNet to achieve high performance with reduced parameter complexity underscores its potential for scalability and broader application across various large-scale data scenarios. For the results of the 1B and 3B models, please refer to Table 9. For the retrieval results, please refer to Figure 5.

## 6.3 Ablation Studies

**Effectiveness of Parameters Sharing.** As discussed in Sec. 5.2, we employ a parameter sharing strategy between decay and key, and the performance comparison is presented in Table 6. The results demonstrate that employing independent parameters for decay and key leads to performance deterioration, highlighting the significance of parameter sharing.

**Effectiveness of MD-TPE.** The proposed MD-TPE provides relative positional information under linear complexity. We thus explore the effectiveness of the MD-TPE across all tasks, shown in Table 1,2,6,8. We can observe that removing MD-TPE results in significant performance degradation, particularly for image generation, which highly depends on the relative position of the image content. Similarly, performance comparison in language modeling tasks also confirms the effectiveness of MD-TPE when reduced to a single dimension.

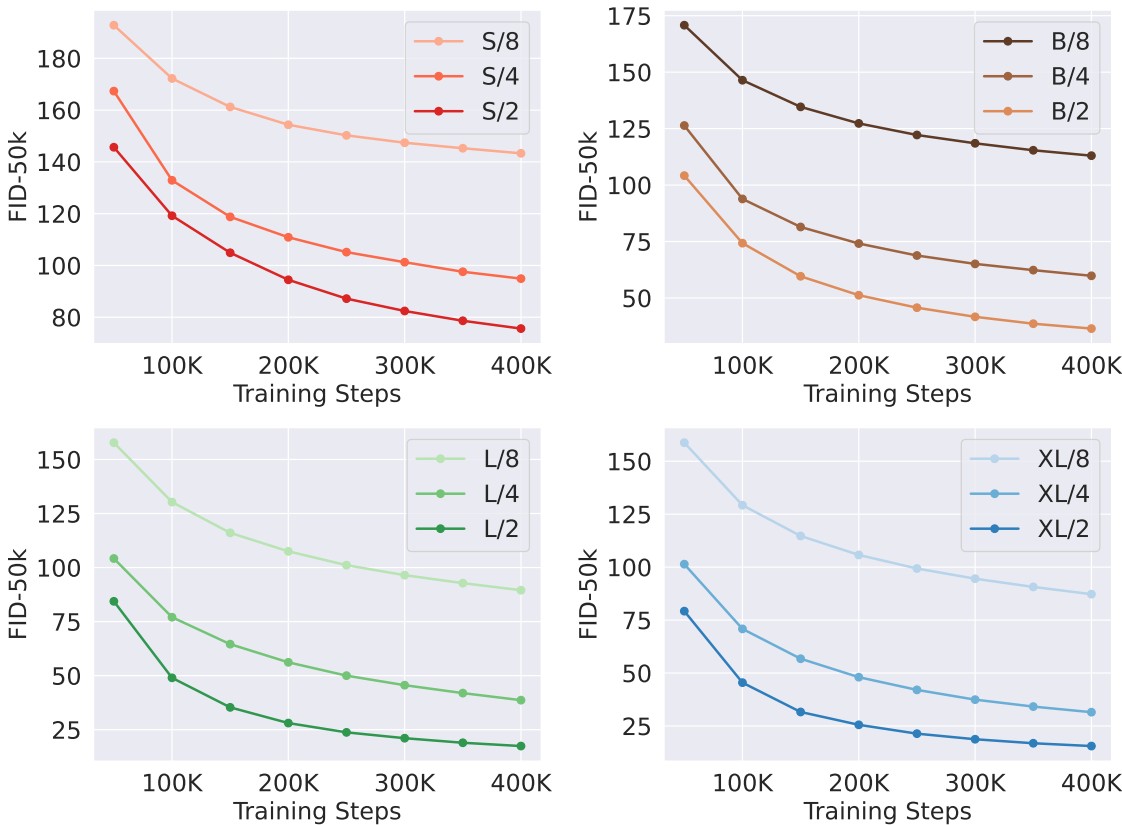

Figure 4: **Scaling up the LightNet enhances the FID during every stages of training.** We present the FID-50K across training iterations for twelve LightNet models. Enhancing the LightNet backbone results in improved generative models for all sizes of models and patches.

Table 7: **Ablation studies on image classification task** on normalization and low-rank output gate.

| Model | Tiny | | Small | | Base | |
|---|---|---|---|---|---|---|
| | Acc (%) ↑ | Params (M) | Acc (%) ↑ | Params (M) | Acc (%) ↑ | Params (M) |
| LightNet | 74.46 | 6.00 | 80.12 | 22.64 | 81.90 | 87.74 |
| LightNet w/o Norm | 73.46 | 6.00 | 79.65 | 22.63 | 81.49 | 87.72 |
| LightNet w full rank output gate | 73.92 | 6.16 | 80.04 | 23.82 | 81.95 | 93.64 |

**Effectiveness of MD-LRPE.** LRPE has already proven its effectiveness in the field of language modeling. Therefore, when faced with higher-dimensional inputs, the contributions of its extension, MD-LRPE should be systematically validated. To this end, we conduct numerous ablation experiments, and the results, as shown in Table 1,2,6,8, demonstrate the effectiveness of extending LRPE into a multi-dimensional space through MD-LRPE operation.

**Normalization and Low Rank output gate.** As shown in Table 7, the performance of the low-rank output gate we used in LightNet is comparable to that of the full-rank output gate, with approximately 5% fewer parameters. Moreover, removing the extra-normalization in LightNet significantly decreases the model's effectiveness.

Table 6: **Performance comparison Pile for large-scale language modeling**. We trained under the 10 billion token subset of Pile, and it can be seen that LightNet's PPL is better than LLaMA's. The best result is highlighted with **bold** and the second with underlined.

| Model | PPL ↓ | Params |
|---|---|---|
| LLaMA | 4.62 | 385M |
| TNL | 4.62 | 379M |
| Mamba | **4.59** | 385M |
| LightNet | **4.59** | 379M |
| LightNet w/o TPE | 4.69 | 379M |
| LightNet w/o LRPE | 4.69 | 379M |
| LightNet no share | 4.76 | 385M |
| LightNet w/o Decay | 4.62 | 379M |

Table 8: **Ablation studies on image generation** for LightNet-B/2 Configurations. We compare the performance of FID under different training steps.

| Model | 50K | 100K | 150K | 200K | 250K | 300K | 350K | 400K |
|---|---|---|---|---|---|---|---|---|
| LightNet-B/2 | 104.19 | 74.27 | 59.60 | 51.22 | 45.70 | 41.65 | 38.60 | 36.45 |
| LightNet-B/2 w/o TPE | 105.86 | 77.64 | 64.81 | 57.24 | 51.98 | 48.12 | 44.90 | 42.74 |
| LightNet-B/2 w/o LRPE | 132.17 | 82.99 | 67.79 | 59.02 | 52.88 | 48.41 | 44.94 | 42.37 |

**Effectiveness of Additive Decay.** We discuss the roles of Additive Decay in the causal setting and the non-causal setting. In Table 1, 6, 9, "LightNet w/o Decay" refers to removing Additive Decay and using the "SiLU" activation function. For the causal setting, we evaluate the impact of removing Additive Decay in language models. The results, as shown in Table 6, 9, reveal that removing Additive Decay decreases the perplexity (PPL) by 0.03. For 1B and 3B parameter language models, removing Additive Decay reduces accuracy on Commonsense Reasoning Tasks by approximately 2%. In the non-causal setting, we test classification performance on ImageNet. As shown in Table 1, removing the Additive Decay (i.e., "Softmax" activation function is this scenery) leads to significant performance degradation across all models. This demonstrates the critical role of the Softmax activation function in non-causal tasks. **Speed Test.** The current linear complexity models employ multiplicative linear recurrence in sequence modeling and necessitate at least two scans for multi-dimensional data, resulting in processing time denoted by the "2 Scan" in Fig. 2. In contrast, our LightNet requires only a single scan, leading to a processing time denoted by the "1 Scan". As evident from the figure, the advantage of the "1 Scan" becomes increasingly pronounced with the growth of sequence length.

## 7  Conclusion

In this paper, we have addressed the inefficiency of "multiplicative decay" linear recurrence in multi-dimensional sequence modeling by introducing a novel "additive decay" linear recurrence that handles multi-dimensional data within a single scan. We developed LightNet, a new multi-dimensional linear attention model enhanced by two new multi-dimensional linear relative positional encoding methods, MD-TPE and MD-LRPE. Empirical evaluations across tasks like image classification, image generation, bidirectional language modeling, and autoregressive language modeling demonstrate LightNet's superior performance and versatility. LightNet offers a significant advancement in efficiency and scalability, providing a promising pathway for future research and applications in multi-dimensional sequence modeling.

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

# A Appendix

## A.1 Proof of Eq 4

Note that

$$
A_t y_t = A_t a_t y_{t-1} + A_t x_t = A_{t-1} y_{t-1} + A_t x_t,
$$
$$
A_t y_t - A_{t-1} y_{t-1} = A_t x_t,
$$
$$
\dots,
$$
$$
A_2 y_2 - A_1 y_1 = A_2 x_2.
$$

By summing up, we can obtain:

$$
A_t y_t - A_1 y_1 = \sum_{s=2}^{t} A_s c x_s, y_t A_t = \sum_{s=1}^{t} A_s x_s, y_t = \sum_{s=1}^{t} \frac{A_s}{A_t} x_s.
$$

## A.2 Further Discussions on Relative Positional Encoding

In this section, we discuss why mainstream relative positional encodings (RPEs) are unsuitable for LightNet. We categorize the main types of RPEs into **Additive RPE** and **Multiplicative RPE** (Qin et al., 2023b) (note that these should not be confused with the "Additive decay" and "Multiplicative decay" linear recurrence discussed in this paper).

**Additive RPE**  Additive RPE (Shaw et al., 2018) is typically expressed in the following form. To simplify the discussion, we omit the scaling factors. Here, $w_{t-s} \in \mathbb{R}$ represents the relative positional encoding:

$$
a_{ts} = \mathbf{q}_t^\top \mathbf{k}_s + w_{t-s}, \mathbf{o}_t = \sum_s \frac{\exp(a_{ts})}{\sum_s \exp(a_{ts})} \mathbf{v}_s.
$$

As shown, Additive RPE requires computation of the attention scores, which is not allowed in LightNet due to compute $\mathbf{K}^\top \mathbf{V}$ first, (Katharopoulos et al., 2020).

**Multiplicative RPE**  The representative work of Multiplicative RPE is RoPE (Su et al., 2021). Although RoPE does not require direct computation of attention scores, it fails to preserve relative positional information when applied to LightNet. Specifically:

$$
\begin{aligned}
\mathbf{o}_t^\top &= \sum_{s \le t} \mathbf{q}_t^\top \mathbf{W}_t^\top \left( \frac{\mathbf{W}_s \exp(\mathbf{k}_s)}{\sum_{j=1}^{t} \exp(\mathbf{k}_j)} \right) \mathbf{v}_s^\top \\
&= \sum_{s \le t} \mathbf{q}_t^\top \mathbf{W}_t^\top \operatorname{diag} \left( \sum_{j=1}^{t} \exp(\mathbf{k}_j) \right)^{-1} (\mathbf{W}_s \exp(\mathbf{k}_s)) \mathbf{v}_s^\top \\
&\neq \sum_{s \le t} \mathbf{q}_t^\top \mathbf{W}_t^\top \mathbf{W}_s \operatorname{diag} \left( \sum_{j=1}^{t} \exp(\mathbf{k}_j) \right)^{-1} \exp(\mathbf{k}_s) \mathbf{v}_s^\top \\
&= \sum_{s \le t} \mathbf{q}_t^\top \mathbf{W}_{t-s}^\top \operatorname{diag} \left( \sum_{j=1}^{t} \exp(\mathbf{k}_j) \right)^{-1} \exp(\mathbf{k}_s) \mathbf{v}_s^\top.
\end{aligned}
$$

Here, $\mathbf{W}_t$ represents the rotation matrix in RoPE. The inequality in the second-to-last step arises because **block-diagonal matrices (RoPE matrices) and diagonal matrices are non-commutative**.

**Why LRPE is Chosen**  The above issue does not arise in LRPE (Qin et al., 2023b), which is implemented as:

$$
f_{\text{lrpe}}(\mathbf{x}_t, \Theta) = \operatorname{concat}([\mathbf{x} \odot \cos(t\Theta), \mathbf{x} \odot \sin(t\Theta)], \dim = -1).
$$

Thus, the computation becomes:

$$\mathbf{o}_t^\top = \sum_{s \le t} [\mathbf{q}_t \odot \cos(t\Theta), \mathbf{q}_t \odot \sin(t\Theta)]^\top \left[ \frac{\exp(\mathbf{k}_s) \odot \cos(s\Theta)}{\sum_{j=1}^t \exp(\mathbf{k}_j)}, \frac{\exp(\mathbf{k}_s) \odot \sin(s\Theta)}{\sum_{j=1}^t \exp(\mathbf{k}_j)} \right] \mathbf{v}_s^\top$$

$$= \sum_{s \le t} \mathbf{q}_t^\top \operatorname{diag}\{\cos((t-s)\Theta)\} \frac{\exp(\mathbf{k}_s)}{\sum_{j=1}^t \exp(\mathbf{k}_j)} \mathbf{v}_s^\top .$$

This demonstrates that LRPE effectively captures relative positional information, making it suitable for LightNet. Hence, we adopt LRPE in our design.

## B    Implementation Details of MD-LRPE

The pseudocode for our implementation is provided below:

```python
def lrpe_cosine_md_torch(x, theta, shape, l=0, act="none", dim=None):
    # x: b, h, n, d; n = l + prod(shape)
    # theta: h, e; e >= round(d + len(shape) - 1) // len(shape))
    # shape: n1, ..., nm
    # l: we do not do lrpe cosine on the first l tokens
    shape = torch.tensor(shape, dtype=torch.int32, device=x.device)
    d = x.shape[-1]
    m = len(shape)

    assert (
        theta.shape[-1] * m >= d
    ), "dim of theta should be larger than round(d + len(shape) - 1) // len(shape)
        )"

    array = [
        torch.arange(n, dtype=torch.int64, device=torch.cuda.current_device())
        for n in shape
    ]
    grid = torch.meshgrid(array)
    index = torch.stack(grid, dim=-1)

    for _ in range(m):
        theta = theta.unsqueeze(1)

    theta_list = []
    for i in range(m):
        theta_list.append(index[..., i : i + 1] * theta.float())

    theta = torch.cat(theta_list, dim=-1)[..., :d]
    theta, ps = pack([theta], "h * d")

    x_no_lrpe = x[:, :, :l]
    x = x[:, :, l:]

    cos = theta.cos()
    sin = theta.sin()

    output = torch.cat([x * cos, x * sin], dim=-1)
    if l > 0:
        output = torch.cat([F.pad(x_no_lrpe, (0, d)), output], dim=-2)

    return output.to(x.dtype)
```

Concretely, we construct a multi-dimensional index. For example, for a ViT token sequence of length

$$n = h \times w, \tag{23}$$

we construct two-dimensional indices

$$(i, j), i \in \{0, \ldots, h-1\}, j \in \{0, \ldots, w-1\}. \tag{24}$$

These indices are multiplied element-wise with $\theta$ and broadcast to produce a per-position parameter matrix

$$\Theta \in \mathbb{R}^{n \times d}, \tag{25}$$

where $d$ denotes the head dimension. We then compute

$$\sin(\Theta), \cos(\Theta), \tag{26}$$

and produce the final output

$$[\mathbf{X} \odot \sin(\Theta), \mathbf{X} \odot \cos(\Theta)]. \tag{27}$$

The overall time complexity is

$$O(nd). \tag{28}$$

## B.1 More experiments

In this section, we provide additional experimental results. In Table 9, we show the performance of LightNet under the Commonsense Reasoning Tasks. In Table 10, we present the advantages of LightNet (1 scan) compared to the 2-scan method. In Table 11, we present the effects of LightNet on image generation tasks across various sizes. In Figure 5, we illustrate the retrieval advantages of LightNet compared to Mamba2.

Table 9: **Performance Comparison on Commonsense Reasoning Tasks.** PS, T, HS, WG stand for parameter size (billion), tokens (billion), HellaSwag, and WinoGrande, respectively.

| Model | P | T | PIQA | HS | WG | ARC-e | ARC-c | OBQA | AVG |
|---|---|---|---|---|---|---|---|---|---|
| OPT | 2.7 | 300 | 73.83 | 60.60 | 61.01 | 60.77 | 31.31 | 35.2.0 | 53.79 |
| Pythia | 2.8 | 300 | 74.10 | 59.31 | 59.91 | 64.14 | 33.02 | 35.60 | 54.35 |
| BLOOM | 3.0 | 350 | 70.57 | 54.53 | 58.48 | 59.43 | 30.38 | 32.20 | 50.93 |
| RWKV-4 | 3.0 | - | 72.42 | 58.75 | 57.30 | 62.92 | 35.15 | 36.20 | 53.79 |
| LightNet | 3.0 | 300 | 75.14 | 60.00 | 59.75 | 65.99 | 33.87 | 35.80 | 55.09 |
| LightNet w/o Decay | 3.0 | 300 | 74.27 | 57.38 | 57.30 | 63.22 | 31.40 | 35.20 | 53.13 |
| LightNet | 1.0 | 300 | 71.06 | 47.27 | 51.30 | 56.31 | 27.56 | 33.00 | 47.75 |
| LightNet w/o Decay | 1.0 | 300 | 70.73 | 45.55 | 50.51 | 55.22 | 27.30 | 31.00 | 46.72 |

Table 10: Performance comparison for image generation task on ImageNet1k, where LightNet use 1 scan, Tnl/RetNet and Hgrn2 use 2 scan.

| Model | 50K | 100K | 150K | 200K | 250K | 300K | 350K | 400K |
|---|---|---|---|---|---|---|---|---|
| LightNet-B/8 | 170.79 | 146.43 | 134.63 | 127.31 | 122.18 | 118.50 | 115.40 | 113.02 |
| Tnl/RetNet-S/8 | 178.96 | 150.09 | 136.36 | 127.92 | 122.77 | 118.92 | 115.64 | 113.36 |
| Hgrn2-S/8 | 182.75 | 152.13 | 140.94 | 133.95 | 129.14 | 125.78 | 123.27 | 121.08 |

## B.2 Configurations

In this section, we provide training configurations for all experiments. The configuration for Bidirectional Language Modeling is the same as (Geiping & Goldstein, 2022), while the configurations for the other experiments are as shown in Table 12, 13, 14, 15. We use Pytorch (Paszke et al., 2019) and A100 for training.

Table 11: Performance Metrics Across Different LightNet Configurations

| Model | 50K | 100K | 150K | 200K | 250K | 300K | 350K | 400K |
|---|---|---|---|---|---|---|---|---|
| LightNet-S/8 | 192.79 | 172.23 | 161.23 | 154.34 | 150.25 | 147.40 | 145.27 | 143.31 |
| LightNet-S/4 | 167.33 | 132.89 | 118.77 | 110.88 | 105.15 | 101.25 | 97.56 | 94.90 |
| LightNet-S/2 | 145.66 | 119.20 | 104.90 | 94.45 | 87.18 | 82.41 | 78.63 | 75.61 |
| DiT-S/2 | - | - | - | - | - | - | - | 67.16 |
| LightNet-B/8 | 170.79 | 146.43 | 134.63 | 127.31 | 122.18 | 118.50 | 115.40 | 113.02 |
| LightNet-B/4 | 126.37 | 93.86 | 81.44 | 74.11 | 68.80 | 65.09 | 62.34 | 59.81 |
| LightNet-B/2 | 104.19 | 74.27 | 59.60 | 51.22 | 45.70 | 41.65 | 38.60 | 36.45 |
| DiT-B/2 | - | - | - | - | - | - | - | 42.76 |
| LightNet-L/8 | 157.76 | 130.29 | 116.06 | 107.50 | 101.10 | 96.47 | 92.79 | 89.51 |
| LightNet-L/4 | 104.18 | 77.02 | 64.55 | 56.16 | 49.99 | 45.58 | 41.91 | 37.54 |
| LightNet-L/2 | 84.38 | 48.98 | 35.32 | 28.05 | 23.75 | 21.06 | 18.94 | 17.42 |
| DiT-L/2 | - | - | - | - | - | - | - | 24.37 |
| LightNet-XL/8 | 158.75 | 129.23 | 114.72 | 105.75 | 99.35 | 94.53 | 90.66 | 87.22 |
| LightNet-XL/4 | 101.39 | 70.84 | 56.75 | 48.04 | 42.04 | 37.43 | 34.16 | 31.51 |
| LightNet-XL/2 | 79.22 | 45.46 | 31.61 | 25.55 | 21.37 | 18.74 | 16.84 | 15.52 |
| DiT-XL/2 | - | - | - | - | - | - | - | 19.20 |

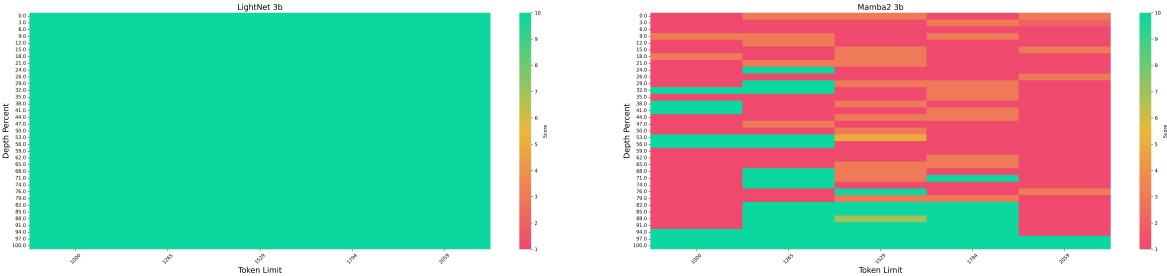

Figure 5: The Needle-in-the-Haystack results of LightNet and Mamba2, all outcomes were evaluated using GPT, where higher scores indicate better performance. Following (Qin et al., 2024b), we used the easy model, as it is more compatible with the base model. The results demonstrate that LightNet outperforms Mamba2.

Table 12: **Comprehensive Configurations of the Model and Training Procedures for LightNet Experiments** "Total batch size" means batch_per_gpu × update_freq × num_gpus; "ALM" stands for Autoregressive Language Model; "IM" stands for Image Modeling, "IG" stands for image generation.

|  | ALM | IM | IG |
|---|---|---|---|
| Dataset | WikiText-103 | ImageNet-1k | ImageNet-1k |
| Tokenizer method | BPE | - | - |
| Src Vocab size | 50265 | - | - |
| Sequence length | 512 | - | - |
| Total batch size | 128 | 2048 | 256 |
| Number of updates/epochs | 50k updates | 300 epochs | 80 epochs |
| Warmup steps/epochs | 4k steps | 20 epochs | - |
| Peak learning rate | 5e-4 | 5e-4 | 1e-4 |
| Learning rate scheduler | Inverse sqrt | Cosine | - |
| Optimizer | Adam | Adamw | Adamw |
| Adam $\epsilon$ | 1e-8 | 1e-8 | 1e-8 |
| Adam $(\beta_1, \beta_2)$ | (0.9, 0.999) | (0.9, 0.98) | (0.9, 0.98) |
| Weight decay | 0.1 | 0.1 for Base, else 0.05 | 0 |
| Gradient clipping | - | 5.0 | - |
| GPUS | 4 | 8 | 8 |

Table 13: **Configurations for LLM**

| Params(B) | Layers | Hidden Dim | L.R. | Batch Size Per GPU | SeqLen | GPUs |
|---|---|---|---|---|---|---|
| 1 | 18 | 2048 | 3.00E-04 | 10 | 2048 | 16 |
| 3 | 36 | 2560 | 3.00E-04 | 36 | 2048 | 48 |

Table 14: **Model Configurations for Image Generation task.**

| Model | Layers | Hidden Dim | Heads | Params |
|---|---|---|---|---|
| LightNet-S | 18 | 384 | 6 | 33M |
| LightNet-B | 18 | 768 | 6 | 131M |
| LightNet-L | 36 | 1024 | 16 | 470M |
| LightNet-XL | 42 | 1152 | 16 | 680M |

Table 15: **Model Configurations for Image Classification task.**

| Model | Layers | Hidden size | Heads | Params |
|---|---|---|---|---|
| LightNet-T | 12 | 192 | 6 | 6.0M |
| LightNet-S | 12 | 384 | 16 | 22.6M |
| LightNet-B | 12 | 768 | 16 | 87.7M |

