# OpenReview forum: "You Only Scan Once: Efficient Multi-dimension Sequential Modeling with LightNet"
_TMLR — Accepted by TMLR_

### Review · Reviewer_9yDA · 2025-06-26

**Summary Of Contributions:**

In this paper the authors touch upon a very interesting topic in the implementation of recurrence that is applicable to transformers and beyond. The authors highlight the problems associated with the inefficiencies caused by multiplicative decay in linear recurrence and "introduce" additive decay in linear recurrence that could handle multi-dimensional data within a single scan. They further use it to develop LightNet, a linear attention model with positional encoding methods (MD-TPE and MD-LRPE). They claim superior performance on image classification, generation and language modeling tasks.

**Audience:**

Yes

**Broader Impact Concerns:**

i believe the work in itself doesn't have any broader impact concerns but might be used in other works that could have a societal impact. If the authors choose to do so, they can highlight it in the manuscript.

**Claims And Evidence:**

Yes

**Requested Changes:**

In figure 3, please expand LNA to LightNet attention since the image could be rendered before the abbreviation is defined in text.
The authors also do not mention about hardware, training setup, time, etc. i saw A100 being used for training in the appendix, but is it a single or multiple? Was everything trained on A100? How long did it take?
The authors should also mention how the model might work when performing classification in an adversarial setting, when bottleneck/attention methods could have similar performance -- look at EITs, Imagenet-C, etc.
The authors claim that they propose additive decay for linear recurrence. The authors are certainly not the first ones there. Can the authors cite relevant work and expand on a bit of a background on how they are different from the previous work(s)?

**Strengths And Weaknesses:**

The manuscript has quite strong mathematical foundations, with the authors highlighting everything using equations where possible. i also like the separate preliminary section to give the bare-minimum insights to the reader about what is happening and what to expect.

The authors mention image classification tasks -- how does the model perform on complex classification tasks like those where the object representation becomes important? Have the authors looked at different transforms like Extreme Image Transforms (EITs) for this? Can authors provide insights into how the model is classifying images -- through saliency maps? How does 1 scan help compared to multi-scan? The same for generation?
In table 1, the results for LightNet are not particularly the best on imagenet. Can the authors explain why this might be the case and does it really give an edge over other models for large scale image classification tasks?
In table 3, LN shows improvements compared to other models -- is it coming from the higher number of params in LN vs other models? Do the authors have a way of quantifying that?

---

### Review · Reviewer_BLVa · 2026-02-23

**Summary Of Contributions:**

This paper introduces **LightNet**, a linear-time attention architecture built upon additive decay linear recurrence. The authors first provide a theoretical analysis of linear recurrence representations, distinguishing multiplicative decay and additive decay forms, and demonstrate that additive decay enables a globally normalizable structure that admits single-pass computation in non-causal settings.

Recognizing that additive decay recurrence is permutation invariant and lacks inherent locality bias, the paper introduces two relative positional encoding schemes tailored for linear attention: **MD-TPE (Multi-Dimensional Toeplitz Positional Encoding)** and **MD-LRPE (Multi-Dimensional Linearized Relative Positional Encoding)**. MD-TPE formulates positional encoding as structured Toeplitz operators parameterized via state-space models to maintain linear scan complexity, while MD-LRPE generalizes phase-based relative encoding to multi-dimensional inputs through feature grouping and complex rotations that preserve relative positional differences in inner products.

Overall, the paper contributes both (i) a theoretical clarification of recurrence structures underlying linear attention and (ii) architectural mechanisms for efficient multi-dimensional, position-aware modeling.

**Audience:**

Yes

**Broader Impact Concerns:**

The work focuses on improving architectural efficiency and positional modeling for neural networks. There are no direct ethical risks specific to this method beyond general concerns associated with scaling large models (e.g., environmental cost of training and potential misuse in generative systems). The broader impact is primarily technical.

**Claims And Evidence:**

Yes

**Requested Changes:**

1. Provide a short discussion clarifying the expressive implications of additive decay, particularly in comparison to multiplicative decay. A theoretical or qualitative explanation would suffice.

2. Clarify the practical and conceptual differences between MD-TPE and MD-LRPE, including when each is preferable. This can be addressed through explanation rather than additional experiments.

3. Clarify implementation details of MD-LRPE, specifically whether complex numbers are explicitly used or implemented via real-valued trigonometric decomposition, and comment on computational overhead.

**Strengths And Weaknesses:**

### Strengths

1. **Clear Theoretical Characterization of Linear Recurrence**
   The distinction between multiplicative and additive decay forms is mathematically clean and clarifies representational constraints of recurrence-based linear attention.

2. **Efficient Additive Decay Reformulation**
   The additive decay structure enables globally normalized aggregation and 1-scan computation in non-causal settings, which is computationally appealing.

3. **Principled Multi-Dimensional Positional Encoding**
   The extension of relative positional encoding to multi-dimensional inputs is technically sound. In particular, the MD-LRPE formulation ensures that
   [
   W_n^H W_m = W_{m-n},
   ]
   preserving relative positional dependence in a mathematically elegant manner.

4. **Computational Awareness**
   The use of SSM parameterization for Toeplitz kernels to achieve linear-time scan complexity demonstrates careful algorithmic design.

### Weakness

1. Expressive trade-off of additive decay is under-discussed. The paper theoretically characterizes additive decay but does not sufficiently discuss its expressive limitations relative to multiplicative decay or softmax attention. While efficiency advantages are clear, the representational trade-offs are not deeply analyzed.

2. Conceptual distinction between MD-TPE and MD-LRPE is not fully clarified. Although both positional encoding schemes are evaluated, the paper does not clearly articulate their inductive differences or provide guidance on when one should be preferred over the other. The ablation tables show performance differences, but the conceptual interpretation remains limited.

3. Practical implementation details of MD-LRPE are insufficient. The MD-LRPE formulation relies on complex-valued phase rotations. The paper does not clearly explain whether the implementation uses real-valued decomposition or complex tensors, nor does it discuss computational overhead or stability considerations.

---

> ### Author Response · Authors · 2026-02-26
>
> ## Q1. Implications of Additive Decay, Particularly in Comparison to Multiplicative Decay
>
> ### A1.
>
> The principal distinction between additive positional encoding and multiplicative positional encoding lies in their ability to represent relative distance. Specifically, additive positional encoding cannot explicitly encode relative distance, whereas multiplicative positional encoding can.
>
> Consider the linear recurrence:
> $$
> y_t = a_t y_{t-1} + x_t, \quad y_0 = 0,
> $$
> with its expanded form:
> $$
> y_t = \sum_{s=1}^t c_{ts} x_s.
> $$
>
> ------
>
> ### Multiplicative Decay
>
> For multiplicative decay, the coefficient $c_{ts}$ can be written as:
> $$
> \begin{aligned}
> c_{ts}
> &= \frac{g_s}{g_t} \\\\
> &= \exp(\log g_s - \log g_t) \\\\
> &= \exp \left( -\sum_{j=s+1}^t a_j \right).
> \end{aligned}
> $$
> Moreover, it satisfies the bounds:
> $$
> \rho_1^{\,t-s} \le c_{ts} \le \rho_2^{\,t-s},
> $$
> where
> $$
> \rho_1 = \min_{j} \exp(-a_j),
> \rho_2 = \max_{j} \exp(-a_j).
> $$
> Thus, the upper and lower bounds of $c_{ts}$ (which can be interpreted analogously to attention scores) are explicit functions of the relative distance $t-s$.
>
> ------
>
> ### Additive Decay
>
> For additive decay, the coefficient is given by:
> $$
> \begin{aligned}
> c_{ts}
> &= \frac{g_s}{g_t} \\\\
> &= \frac{\sum_{i=1}^s \delta_i}{\sum_{i=1}^t \delta_i}.
> \end{aligned}
> $$
> It follows that:
> $$
> \frac{s \mu_1}{t \mu_2}
> \le c_{ts}
> \le
> \frac{s \mu_2}{t \mu_1},
> $$
> where
> $$
> \mu_1 = \min_{j} \delta_j,
> \mu_2 = \max_{j} \delta_j.
> $$
> In contrast to the multiplicative case, the bounds of $c_{ts}$ under additive decay depend on the absolute indices $s$ and $t$, rather than solely on their relative distance $t-s$.
>
> ------
>
> ### Implication
>
> Therefore, multiplicative decay yields coefficients whose bounds are explicit functions of relative distance, while additive decay does not. This fundamental difference motivates the introduction of MD-TPE and MD-LRPE in our subsequent design, as they enable explicit modeling of relative positional structure.
>
> ## Q2. Practical and Conceptual Differences Between MD-TPE and MD-LRPE
>
> ### A2.
>
> From a practical standpoint, MD-TPE is applied only once, immediately after the embedding layer. In contrast, MD-LRPE is incorporated into every attention module and is therefore applied as many times as the number of transformer layers.
>
> Consequently, MD-TPE incurs lower computational overhead than MD-LRPE. In scenarios where maximizing computational efficiency is critical, we recommend using MD-TPE exclusively.
>
> From an inductive bias perspective, MD-TPE introduces relative positional information via exponential decay, thereby imposing a strong locality prior. In contrast, MD-LRPE does not rely on exponential decay and thus embodies a comparatively weaker locality prior.
>
> ------
>
> ### Summary
>
> 1. If computational efficiency is the primary concern, we recommend using MD-TPE alone.
> 2. If the underlying data exhibits a strong locality prior, MD-TPE is preferable; otherwise, MD-LRPE may be more suitable.
>
> ## Q3. Implementation Details of MD-LRPE
>
> ### A3.
>
> The pseudocode for our implementation is provided below:
>
> ```
> def lrpe_cosine_md_torch(x, theta, shape, l=0, act="none", dim=None):
>     # x: b, h, n, d; n = l + prod(shape)
>     # theta: h, e; e >= round(d + len(shape) - 1) // len(shape))
>     # shape: n1, ... , nm
>     # l: we do not do lrpe cosine on the first l tokens
>     shape = torch.tensor(shape, dtype=torch.int32, device=x.device)
>     d = x.shape[-1]
>     m = len(shape)
>
>     assert (
>         theta.shape[-1] * m >= d
>     ), "dim of theta should be larger than round(d + len(shape) - 1) // len(shape))"
>
>     array = [
>         torch.arange(n, dtype=torch.int64, device=torch.cuda.current_device())
>         for n in shape
>     ]
>     grid = torch.meshgrid(array)
>     index = torch.stack(grid, dim=-1)
>
>     for _ in range(m):
>         theta = theta.unsqueeze(1)
>
>     theta_list = []
>     for i in range(m):
>         theta_list.append(index[..., i : i + 1] * theta.float())
>
>     theta = torch.cat(theta_list, dim=-1)[..., :d]
>     theta, ps = pack([theta], "h * d")
>
>     x_no_lrpe = x[:, :, :l]
>     x = x[:, :, l:]
>
>     cos = theta.cos()
>     sin = theta.sin()
>
>     output = torch.cat([x * cos, x * sin], dim=-1)
>     if l > 0:
>         output = torch.cat([F.pad(x_no_lrpe, (0, d)), output], dim=-2)
>
>     return output.to(x.dtype)
> ```
>
> Concretely, we construct a multi-dimensional index. For example, for a ViT token sequence of length
> $
> n = h \times w,
> $
> we construct 2D indices:
> $$
> (i, j),
> \quad
> i \in \{0, \ldots, h-1\},
> \quad
> j \in \{0, \ldots, w-1\}.
> $$
> These indices are multiplied element-wise with $\theta$ and broadcast to produce a per-position parameter matrix:
> $$
> \Theta \in \mathbb{R}^{n \times d},
> $$
> where $d$ denotes the head dimension.
>
> We then compute:
> $
> \sin(\Theta),  \cos(\Theta),
> $
> and produce the final output:
> $
> \left[
> \mathbf{X} \odot \sin(\Theta),
> \;
> \mathbf{X} \odot \cos(\Theta)
> \right].
> $
> The overall time complexity is:
> $$
> O(nd).
> $$

---

### Review · Reviewer_NA8C · 2026-03-01

**Summary Of Contributions:**

The paper proposes LightNet, a linear attention layer for multi-dimensional sequence modeling. The core contribution is replacing the standard linear attention recurrence with an **additive decay** variant. In standard causal gated linear attention, the state update is:

$$kv_t = kv_{t-1} + \text{SiLU}(k_t) v_t^\top, \quad o_t = \text{Norm}[kv_t^\top \cdot \text{SiLU}(q_t)] \odot \text{Sigmoid}(u_t)$$

where $kv_t \in \mathbb{R}^{d \times d}$ is the state at timestep $t$ and $k_t, v_t, q_t \in \mathbb{R}^d$.

The authors propose replacing the state recurrence with:

$$s_t = s_{t-1} + \exp(k_t), \quad \bar{k}_t = \exp(k_t)/s_t$$
$$kv_t = \text{diag}\{1 - \bar{k}_t\}kv_{t-1} + \bar{k}_t v_t^\top$$

where $\bar{k}_t \in \mathbb{R}^d$ tracks the channel-wise cumulative softmax. Unrolling the recurrence gives:

$$kv_t = \frac{1}{s_t}\sum_{\tau=1}^{t} \exp(k_\tau) v_\tau^\top$$

The primary change over the no-decay baseline is therefore a **running normalization** of the accumulated state.

For the **non-causal (bidirectional) case**, the formulation reduces to:

$$O = \text{Norm}\left[\text{SiLU}(Q)\left(\text{Softmax}(K)^\top V\right)\right] \odot \text{Sigmoid}(U)$$

which, as noted by Katharopoulos et al. (2020), is a standard matrix multiplication with no sequential recurrence and  significantly faster than any scan-based approach. Since this formulation is permutation-invariant, positional embeddings must be incorporated explicitly. The paper proposes two methods for this:

1. **MD-TPE:** positional information is injected via a channel-wise 1D diagonal SSM scan (Gu et al., 2022) applied independently along each spatial dimension (e.g., height and width for images) and summed. This is applied once at the bottom of the network, similar to SPADE (Zuo et al., 2022).

2. **MD-LRPE:** injects relative positional information for each spatial dimension independently. Given a query or key vector $x \in \mathbb{R}^d$ at position $(n_1, \ldots, n_k)$, the $d$ features are partitioned into $k$ groups of size $d/k$ (one per spatial dimension). The $s$-th group is scaled as:
$$x_j \leftarrow x_j \cdot \exp(in_s\theta_j), \quad \frac{sd}{k} < j \leq \frac{(s+1)d}{k}, \quad \theta_j = 10000^{-2j/d}$$
For a 2D image ($k=2$), the first $d/2$ features are rotated by row position $n_1$ and the last $d/2$ by column position $n_2$.

Experiments span image classification, image generation, bidirectional language modeling (GLUE), and autoregressive language modeling (WikiText-103, Pile, 1B/3B commonsense reasoning).

**Audience:**

Yes

**Broader Impact Concerns:**

No significant ethical concerns. The work presents a general-purpose sequence modeling architecture. No Broader Impact Statement required.

**Claims And Evidence:**

Yes

**Requested Changes:**

1. Provide a more explicit comparison of the non-causal LightNet formulation to Katharopoulos et al. (2020) with softmax-normalized keys, clarifying precisely what is novel. Include a LightNet ablation without softmax.

2. The gains over the no-decay baseline are small in most settings (0.03 PPL on Pile, 0.17% for Small ImageNet). The authors should temper their claims about the necessity of additive decay.

3. The efficiency advantage of the proposed approach is clear in the non-causal case (matrix multiplication vs. scan). However, in the causal case, the speedup over existing approaches is unclear. Many competing methods introduce additional complexity specifically to improve generalization to longer sequences - an experiment in this setting would be helpful.

**Strengths And Weaknesses:**

### Strengths

1. **Simplicity.** The proposed formulation - running normalization of the key-value state in the causal case, and plain matrix multiplication in the non-causal case — works as well as more complex multi-scan approaches. The performance differences over the no-decay baseline are modest (0.03 PPL on Pile, 0.17% accuracy for Small ImageNet), suggesting that this even simpler formulation is an attractive alternative. This is also a useful empirical finding.

2. **Comprehensive and broad experiments.** The evaluation spans image classification, image generation, bidirectional language modeling, and autoregressive language modeling at scales up to 3B parameters. This breadth makes the results more convincing than typical single-domain evaluations.

3. **Theoretical justification.** The paper provides formal grounding via Theorem 4.1 (characterizing representable linear combinations) as well as a clear explanation in Appendix A.2 of why RoPE fails in this setting and why LRPE is compatible.


### Weaknesses

1. **Modest novelty over existing methods.** The non-causal formulation reduces to Katharopoulos et al. (2020) with softmax-normalized keys. The causal formulation's primary change over the no-decay baseline is a running normalization of the key-value state, closely related to the denominator term discussed in prior linear attention work (Qin et al., 2022). The positional encoding methods (MD-TPE, MD-LRPE) are natural and expected extensions of existing 1D methods (SSM scans, LRPE) to multiple dimensions via separable decomposition.

2. **Significant related work with similar formulations.** The proposed formulation is closely related to several concurrent and prior works. The running softmax normalization in the causal state update resembles GLA (Yang et al., 2023). The separable per-axis SSM scan in MD-TPE is similar to applying Mamba or S4 independently per spatial axis, as explored in related vision SSM works. The density of related work with overlapping formulations makes it difficult to clearly delineate the specific contribution of this paper.

---

> ### Author Response · Authors · 2026-03-02
> **Response to Reviewer NA8C**
>
> **Q1. Provide a more explicit comparison of the non-causal LightNet formulation to Katharopoulos et al. (2020) with softmax-normalized keys, clarifying precisely what is novel. Include a LightNet ablation without softmax.**
>
> **A1.** Thank you for this question. In the non-causal setting, LightNet is indeed equivalent to applying a softmax activation over the keys along the sequence dimension. The main contribution of our work is not the non-causal formulation in isolation, but rather the observation that the need for multiple scan operations in prior methods stems from the particular design of the decay mechanism. We show that this issue can be resolved through an additive decay formulation, which leads to a unified framework covering both the causal and non-causal cases. In the non-causal case, this formulation reduces to applying softmax over the keys along the sequence dimension, which also provides theoretical insight into why such a softmax normalization is reasonable.
>
> Regarding the requested ablation without softmax, this result is already reflected in the last row of Table 1, labeled **“LightNet w/o Decay.”** As shown there, removing the decay/normalization mechanism leads to a clear degradation in performance compared to the full LightNet model.
>
> ---
>
> **Q2. The gains over the no-decay baseline are small in most settings (0.03 PPL on Pile, 0.17% for Small ImageNet). The authors should temper their claims about the necessity of additive decay.**
>
> **A2.** Thank you for this helpful suggestion. We agree that our current wording may overstate the importance of additive decay. In the revised version, we will moderate our claims and describe the results more precisely, namely that additive decay performs slightly better than the no-decay variant across most settings.
>
> ---
>
> **Q3. The efficiency advantage of the proposed approach is clear in the non-causal case (matrix multiplication vs. scan). However, in the causal case, the speedup over existing approaches is unclear.**
>
> **A3.** We thank the reviewer for raising this point. To better evaluate efficiency in the causal setting, we compared the throughput of LightNet against GLA, HGRN2, and Mamba2 on causal language modeling. All models have approximately 1.45B parameters and were evaluated on 8×A100 GPUs. The results are shown below:
>
> | Model | Params | TGS (Tokens per GPU per Second) |
> | --- | --- | --- |
> | HGRN2 | 1.45B | 18386.1 |
> | GLA | 1.45B | 18354.5 |
> | Mamba2 | 1.45B | 18428.3 |
> | LightNet | 1.45B | 16992.2 |
>
> As shown in the table, LightNet is approximately 7% slower than these competing methods in the causal setting. We will include this result in the revision and clarify that, unlike in the non-causal case where LightNet enjoys a clear efficiency advantage, its speed in the causal case is slightly lower than that of several highly optimized alternatives.

---

### Author Response · Authors · 2026-05-09
**The camera-ready version has been uploaded.**

Dear Reviewers and Action Editors,

Thank you for your suggestions on our paper. The camera-ready version has been uploaded. We have incorporated the reviewers’ comments into the paper and marked the changes in blue font.

---

> ### Comment · Action_Editor_dsCb · 2026-05-14
> **Addressing reviewer comments**
>
> Hi,
> I looked at the changes a few remarks:
> 1. seems like there is some problem referencing table 5 from the text (it says table 6.1 -- top of page 11)
> 2. In the paragraph "Efficiency benchmarks" please add a sentence discussing the results explicitly saying that lightnet is similar but a bit behind the alternatives in the causal case.
> 3. Similarly, and as explicitly asked by reviewer NA8C and myself -- please add a sentence in the paragraph somewhere that explicitly and clearly says that efficiency advantages are in the non-causal case, but in the causal case we see similar throughput.

---

> > ### Comment · Editors_In_Chief · 2026-05-25
> >
> > Checking in on this, authors.

---

> > > ### Author Response · Authors · 2026-05-26
> > > **Response to Action Editors**
> > >
> > > Dear Action Editors,
> > >
> > > Apologies for the delayed response. We will update the manuscript in the coming days based on your suggestions.

---

> ### Author Response · Authors · 2026-06-01
> **The manuscript has been updated**
>
> Dear Action Editors,
>
> We have revised the manuscript in accordance with your comments. Please feel free to let us know if you have any further questions or concerns.

---

> > ### Comment · Action_Editor_dsCb · 2026-06-01
> > **Thank you.**
> >
> > I am really sorry but I made an error in my 3rd request above, I meant to write:
> >
> > please add a sentence in the **introduction** somewhere that explicitly and clearly says that efficiency advantages are in the non-causal case, but in the causal case we see similar throughput.
> >
> > So please do that, and upload a new version without the blue font, so that I can approve the final version (after I click that there is no changing of the pdf without PC intervention)
> >
> > Thanks for the great work!

---

> ### Author Response · Authors · 2026-06-01
> **The manuscript has been updated**
>
> We have added a description of the speed performance in the Introduction section: “LightNet achieves a speed advantage in non-causal settings while maintaining comparable speed in causal settings.”

---

### Decision · Action_Editor_dsCb · 2026-04-10

**Recommendation:** Accept with minor revision

**Additional Comments:**

Fruitful discussion took place between authors and reviewers, where reviewers raised several important points about relation to past work, efficiency in the causal setup, implementation details, and more precise phrasing of the precise contribution of this work compared to prior work.

Please follow the requested changes from the reviewers including the new results reported during the rebuttal. In particular follow the changes proposed by reviewers NA8C and BLVa. In your resubmission please make sure to mark where those changes were addressed as I will check indeed they were taken into account.

**Audience:**

Yes

**Audience Explanation:**

Developing efficient mechanisms for linear attention models is of high interest and this paper investigates this in the multi-dimensional setup.

**Claims And Evidence:**

Yes

**Claims Explanation:**

The paper proposes an alternative decay linear recurrence that allow for much more efficient processing in the non-causal case, with broad empirical evaluation and good theory. The non-causal formulation is related to past work but indeed this work provides a good framework with nice empirical results.